**Data Availability Statement:** National Research Center for Cardiac Surgery" Institutional Data Access (contact via phone or mail) for researchers

# Mortality predictors of hospitalized patients with COVID-19: Retrospective cohort study from Nur-Sultan, Kazakhstan

Yuriy Pya[1], Makhabbat Bekbossynova[2], Abduzhappar Gaipov[3], Timur Lesbekov[1], Timur Kapyshev[4], Aidyn Kuanyshbek[5], Ainur Tauekelova[2], Liya Litvinova[6], Aliya Sailybayeva👤[7]*, Ivan Vakhrushev[5], Antonio Sarria-Santamera[3]

1 Department of Cardiac Surgery, National Research Cardiac Surgery Center, Nur-Sultan, Kazakhstan, 2 Department of Cardiology, National Research Cardiac Surgery Center, Nur-Sultan, Kazakhstan, 3 Department of Medicine, School of Medicine, Nazarbayev University, Nur-Sultan, Kazakhstan, 4 Center of Excellence, National Research Cardiac Surgery Center, Nur-Sultan, Kazakhstan, 5 Department of Anesthesiology, Reanimation and Intensive Care Unit, National Research Cardiac Surgery Center, Nur-Sultan, Kazakhstan, 6 Quality and Risk Management Department, National Research Cardiac Surgery Center, Nur-Sultan, Kazakhstan, 7 Research Department, National Research Cardiac Surgery Center, Nur-Sultan, Kazakhstan

* cardiacsurgeryres@gmail.com

## Abstract

### Background

First reported case of Severe acute respiratory syndrome-related coronavirus 2 (SARS-CoV-2) in Kazakhstan was identified in March 2020. Many specialized tertiary hospitals in Kazakhstan including National Research Cardiac Surgery Center (NRCSC) were re-organized to accept coronavirus disease 2019 (COVID-19) infected patients during summer months of 2020. Although many studies from worldwide reported their experience in treating patients with COVID-19, there are limited data available from the Central Asia countries. The aim of this study is to identify predictors of mortality associated with COVID-19 in NRCSC tertiary hospital in Nur-Sultan, Kazakhstan.

### Methods

This is a retrospective cohort study of patients admitted to the NRCSC between June 1st– August 31st 2020 with COVID-19. Demographic, clinical and laboratory data were collected from electronic records. In-hospital mortality was assessed as an outcome. Patients were followed-up until in-hospital death or discharge from the hospital. Descriptive statistics and factors associated with mortality were assessed using univariate and multivariate logistic regression models.

### Results

Two hundred thirty—nine admissions were recorded during the follow-up period. Mean age was 57 years and 61% were males. Median duration of stay at the hospital was 8 days and 34 (14%) patients died during the hospitalization. Non-survivors were more likely to be admitted later from the disease onset, with higher fever, lower oxygen saturation and

who meet the criteria for access to confidential data. The data underlying the results presented in the study are available from (Non-commercial joint-stock company "National Research Center for Cardiac Surgery", research department, phone number: +77172703153, mail: cardiacsurgeryres@gmail.com).

**Funding:** This research has been funded by the Science Committee of the Ministry of Education and Science of the Republic of Kazakhstan (Grant No. BR10965164). https://www.ncste.kz/en/main. The funders had no role in study design, data collection and analysis, decision to publish, or preparation of the manuscript.

**Competing interests:** The authors have declared that no competing interests exist.

increased respiratory rate compared to survivors. Leukocytosis, lymphopenia, anemia, elevated liver and kidney function tests, hypoproteinemia, elevated inflammatory markers (C-reactive protein (CRP), ferritin, and lactate dehydrogenase (LDH)) and coagulation tests (fibrinogen, D-dimer, international normalized ratio (INR), and activated partial thromboplastin time (aPTT)) at admission were associated with mortality. Age (OR 1.2, CI:1.01–1.43), respiratory rate (OR 1.38, CI: 1.07–1.77), and CRP (OR 1.39, CI: 1.04–1.87) were determined to be independent predictors of mortality.

## Conclusion

This study describes 14% mortality rate from COVID-19 in the tertiary hospital. Many abnormal clinical and laboratory variables at admission were associated with poor outcome. Age, respiratory rate and CRP were found to be independent predictors of mortality. Our finding would help healthcare providers to predict the risk factors associated with high risk of mortality. Further investigations involving large cohorts should be provided to support our findings.

## Introduction

The first case of pneumonia of unknown etiology was reported in Wuhan, China in December 2019 [1]. On 11th of February, 2020 World Health Organization (WHO) named the disease coronavirus disease 2019 (COVID-19) and it was identified to be caused by the virus from coronavirus family—SARS-CoV-2 [1]. Rapid spread of the disease over first months of 2020 led to the escalation of the situation and in March 2020 WHO has declared a worldwide pandemic caused by SARS-CoV-2 [1].

The first registered case of COVID-19 in Kazakhstan was identified on March 13th 2020 followed by an introduction of a country-wide state of emergency on March 16th 2020 [2,3]. During that time Kazakhstani healthcare facilities have suddenly been overwhelmed with COVID-19 cases, which resulted in critical filling of infectious hospitals with patients all over Kazakhstan [3]. This is why during the peak of the pandemic many specialized tertiary hospitals were immediately transferred to the infectious diseases profile. Often the most severe cases and critically ill patients were admitted to these hospitals, which have sufficient expertise in advanced intensive care unit (ICU care). One of such hospitals was the National Research Cardiac Surgery Center (NRCSC)–a tertiary-level, highly specialized hospital in Nur-Sultan, Kazakhstan. From June to August 2020 the center has been closed for new cardiac admissions and focused solely on treatment of COVID-19 patients.

From the very beginning of the pandemic scientific and medical communities were trying to identify demographic, clinical and laboratory peculiarities that could help to predict the course and outcome and guide treatment. It has been proposed that older age, male gender and number of comorbidities could increase the risk for severe course and poor outcome of the disease [4,5].

Many clinical and laboratory markers have been studied to date [5]. Preliminary studies showed that the level of pro-inflammatory cytokines could correlate with the most severe forms of COVID-19 leading to acute respiratory distress syndrome (ARDS) [5]. Current clinical guides suggest using D-dimers and lactate dehydrogenase (LDH) as possible markers to identify patients at risk of deadly complications [5,6]. In addition, some commonly assessed hematological parameters including white blood cell count (WBC), lymphocyte count and C-

reactive protein (CRP) were proposed as markers for disease severity and poor outcome [5]. Moreover, clinical findings could also correlate with the course of the disease. Period from disease onset to hospitalization, degree of lung involvement based on computerized tomography (CT) could also play an important role in determining the risk for severe disease course [5].

In the face of the COVID-19 pandemic, a number of tertiary hospitals around the globe had to re-organize into the COVID-centers to help with the overflow in the infectious disease centers [6–8]. Similarly, in Kazakhstan, during the peak of pandemic various hospitals were re-organized into COVID centers. One of them was the National Research Cardiac Surgery Center. Often the most severe cases and critically ill patients were referred to our hospital, which has sufficient experience and equipment to provide advanced floor and ICU care. This could explain why the case fatality rate in our study 3–4 folds was higher than nationwide or worldwide COVID-19 fatality rate of 3–4% [2,9].

Indeed, demographical, clinical and laboratory data obtained from patients at admission are of particular interest as independent predictors of poor outcome, since they can be easily obtained and used to triage patients for further distribution into highly specialized facilities for appropriate care. However, the greatest part of knowledge on this topic comes from the developed world, while data from developing countries is still lacking. It is important to compare the local results with worldwide experience. Therefore, the aim of this retrospective cohort study is to identify demographic, clinical and laboratory predictors of mortality of patients with COVID-19 admitted to the tertiary hospital during June-August of 2020 using detailed patient-level clinical information.

## Methods

### Study design and participants

This is a retrospective cohort study, conducted at the National Research Cardiac Surgery Center (NRCSC)–a tertiary highly specialized hospital in Nur-Sultan, Kazakhstan. All 239 patients admitted to the hospital with COVID-19 disease during the period of June 16th–October 16th 2020 were included in the study. No patients were excluded from the study. Patients were admitted through the emergency department. Diagnosis of COVID-19 was defined as either a positive nasopharyngeal swab for SARS-CoV-2 by reverse transcriptase polymerase chain reaction (PCR) and/or clinically supported evidence of SARS-CoV-2 in the absence of polymerase chain reaction (PCR) confirmed test. Nasopharyngeal swabs were collected at the time of admission and tested with RT-PCR assay. Clinical diagnosis was based on signs and symptoms along with radiological findings suggestive of COVID-19 pneumonia. Laboratory and radiological tests were performed in accordance with the national clinical guidelines for treatment of COVID-19 patients. Patients were managed with supportive care and specific pharmacological therapies in accordance with national and hospital protocols for management of COVID-19 guided by the Ministry of Health of the Republic of Kazakhstan.

### Data collection

Demographic data and details of medical history at baseline were collected from hospital electronic records. These characteristics included age, gender, body mass index (BMI), day of disease from initial symptoms to admission, oxygen saturation, cough, temperature, respiratory rate, disease severity, history of hypertension, diabetes, cardiovascular disease, previous cardiac surgery and results of lung computed tomography.

All laboratory data were collected and measured at the baseline during the first day of admission and included but not limited to complete blood counts (CBC) glycated hemoglobin (HbA1c), serum urea, creatinine, alanine transaminase (ALT), aspartate transaminase (AST),

total bilirubin, total protein, albumin, lactate dehydrogenase (LDH), C-reactive protein (CRP), D-dimer, fibrinogen, International normalized ratio (INR), Activated partial thromboplastin time (aPTT), ferritin. Data on in-hospital complications like acute kidney injury (AKI) as well as provided treatment such as supplemental oxygen, mechanical ventilation, renal replacement therapy, extracorporeal membrane oxygenation (ECMO), antiviral (Favirapin, Lopinovir/ Ritonavir, Remdesivir), antibiotics (Cefuroxim, Cefazolin, Ceftriaxon, Cefepim, Moxifloxacin, Aztreonam, Meropinem, Levafloxacin), antimicrobial drug (Fluconazole) anticoagulation and steroids were recorded.

## Ethical statement

The study was approved by the Institutional Review Ethics Committee of the National Research Cardiac Surgery Center (#01-97/2021 from 22/04/21) with exemption from informed consent. There were no known risks to participants expected. During the data collection, all personal information of patients is encoded and the data is depersonalized, so we protect the rights of patients and do not disclose their personal information. Researchers received the electronic database only with information about demographic and clinical characteristics of patients, which was analyzed and reported only in aggregated form, further assuring confidentiality of data.

## Study definitions

Severity of the disease was evaluated based on clinical symptoms in accordance with national clinical guidelines (7). The disease severity was stratified as: *mild disease*—body temperature <38˚C, heart rate <90 bpm, respiratory rate <24/min; *moderate disease*—body temperature 38.1–39˚C, heart rate 90–120 bpm, respiratory rate > 24/min; and *severe disease*—high temperature (over 39˚C) with severe symptoms (headache, myalgia, nausea), heart rate > 120 bpm, respiratory rate >28/min. Acute kidney injury was ascertained using the Kidney Disease Improving Global Outcomes (KDIGO) criteria (8). The respiratory failure was stratified as: *stage 1*—dyspnea, restlessness, RR 25–30 breaths per minute (bpm), $PaO_2$ 70 mmHg, $PaCO_2$ 50 mmHg; *stage 2*—cyanosis, RR 35–40 bpm, tachycardia, elevated BP, $PaO_2$ 60 mmHg, $PaCO_2$ 60 mmHg; and *stage 3*—hypoxic coma, dilated pupils, shock, arrhythmia. Stages of lung involvement based on CT scan also classified as: *CT stage 1* <25%, *CT stage 2*—25–50%, *CT stage 3* 50–75%, and *CT stage 4* > 75% as defined by the national guideline for treatment of COVID-19 patients in adults (7).

## Outcome variable

All of the patients were followed-up from admission to either until discharge from the hospital or in-hospital death. The outcome of interest was in-hospital mortality and associate factors.

## Statistical analysis

Descriptive data are presented as percentages for categorical variables and as mean ± standard deviation (SD) or median (inter quartile range—IQR), as appropriate. All cohorts were divided into two groups as survivors and deceased patients. Categorical variables were compared using χ2 tests. Continuous variables were compared using t tests or Mann–Whitney U tests. The association between demographics, clinical, laboratory, treatment related variable and risk of in-hospital death were examined using unadjusted (univariate) and adjusted (multivariate) logistic regression analyses. The following potential confounders were included in the multi-variable adjusted model based on theoretical considerations based on previous results from the

literature as well as the statistical significance level of univariate analysis: Model 1: age, sex, disease duration at admission, respiratory rate at admission; Model 2: in addition to variables from model 1 included laboratory data as WBC, serum creatinine, total protein, fibrinogen, hemoglobin; Model 3: in addition to variables from model 2 included antivirals and corticosteroid medications. Because of many missing data, those variables which are less than 200 observations (>15% missing data) were not included to the regression analysis.

P values are two-sided and reported as significant at <0.05 for all analyses. All analyses were conducted using STATA IC Version 15.1 (STATA Corporation, College Station, TX).

## Results

### Baseline characteristics

There were 205 patients (85.8%) who survived the disease and 34 patients (14.2%) died during hospitalization. All demographic and baseline clinical and laboratory characteristics of the cohort are summarized in Table 1. One hundred fifty—one patients (63.2%) had PCR confirmed COVID-19 diagnosis. Mean age was 57.3±12.7 years, 61.1% were males (Fig 1). Median length from disease onset to admission was 10 days (IQR 7–14) and non-survivors were admitted to the hospital later than survivors from the beginning of the disease onset. Moreover, at the time of admission non-survivors had lower oxygen saturation, elevated temperature and respiratory rate, (Figs 2–4) higher grade of disease severity, higher stage of respiratory failure and greater degree of lung damage on CT compared to survivors.

The non-survivors had significantly abnormal complete blood count (CBC) parameters, elevated serum creatinine, urea, aspartate aminotransferase (AST), total bilirubin, total protein and albumin. Similarly, in coagulation profile elevated levels of D-dimer, fibrinogen, international normalized ratio (INR) and activated partial thromboplastin time (aPTT) were more often found among deceased patients compared to survivors. Similar trend was observed for inflammatory markers (ferritin, CRP).

Non-survivors were more likely to need oxygen supplementation, invasive mechanical ventilation, ECMO, to receive antivirals, steroids, aspirin and oral anticoagulation as well as develop acute kidney injury (AKI) compared to survivors.

### Association between demographic and clinical data with survival

The crude association between clinical-demographic parameters and risk of death using univariate logistic regression results are summarized in Table 2. Each incremental increase by 5 years of age is associated with 17% higher odds of death (Odds ratio—OR 1.17; 95% confidence interval (CI) 1.01–1.36, p-value (p) = 0.039). Patients with un-controlled diabetes (HbA1c>6.5%) had almost 4-fold higher odds of death (OR 3.96; 95% CI 1.02–15.48; p = 0.048) compared to those, whose HbA1c level was under 6.5%. Each incremental increase by 5 days of later admission is associated with 32% higher risk of death (OR 1.32; 95% CI 1.03–1.68; p = 0.026). Elevated temperature at admission is associated with 2-fold increase odds of death (OR 2.06; 95% CI 1.10–3.83; p = 0.023), each incremental increase by 2 degrees Celsius increase odds of death 4 times. Elevated respiratory rate (RR) (OR 1.23; 95% CI 1.11–1.37; p<0.0001) and reduced oxygen saturation (OR 1.66; 95% CI 1.39–1.99; p<0.0001) also increase odds of death. Each incremental increased of RR by 7 breath/min and each incremental decrease of oxygen saturation by 7% are associated with 4-fold and 3.5-fold risk of death, respectively. Leukocytosis, lymphopenia, anemia, elevated liver and kidney function tests, hypoproteinemia, elevated inflammatory markers (C-reactive protein, ferritin, and LDH) and coagulation tests (fibrinogen, D-dimer, INR, and aPTT) at admission were also associated with higher odds of death in univariate logistic regression analysis (Table 2).

**Table 1. General characteristics of admitted COVID-19 patients.**

| Characteristic | Total (n = 239) | Survived (n = 205) | Deceased (n = 34) | p value |
|---|---|---|---|---|
| Age, year (mean ± SD) | 57.3±12.7 | 56.6±12.96 | 61.5±10.5 | 0.037 |
| Gender, n (%) | | | | 0.047 |
| Male, | 146 (61.1) | 120 (58.6) | 26 (76.5) | |
| Female, | 93 (38.9) | 85 (41.5) | 8 (23.5)_ | |
| BMI (mean ± SD) | 28.9±5.01 | 28.7±5.2 | 30.2±3.3 | 0.15 |
| Obesity stages, n (%) | | | | 0.085 |
| Under-weight | 2 (0.4) | 2 (1.1) | 0 (0) | |
| Normal-weight | 38 (17.2) | 38 (20.3) | 0 (0) | |
| Over-weight | 92 (43.0) | 79 (42.3) | 13 (52.00) | |
| Obese | 80 (37.4) | 68 (36.4) | 12 (48) | |
| **Clinical findings at admission** | | | | |
| Day of disease, days (median [IQR]) | 10 [7–14] | 10 [7–14] | 14 [9.5–18.5] | 0.0048 |
| Saturation, %/100 (mean ± SD) | 89±10 | 92±6 | 76±15 | <0.0001 |
| Cough, n (%) | 130 (54.4) | 114 (55.6) | 16 (47.1) | 0.35 |
| Temperature, ˚C (mean ± SD) | 37.97±0.78 | 37.92±0.77 | 38.32±0.76 | 0.022 |
| Respiratory rate, breath/min (mean ± SD) | 21.6±4.1 | 21.28±3.91 | 26.3±3.9 | <0.0001 |
| Severity stage, n (%): | | | | <0.0001 |
| Mild | 16 (6.7) | 16 (7.8) | 0 (0) | |
| Moderate | 178 (74.5) | 178 (86.8) | 0 (0) | |
| Severe | 45 (18.8) | 11 (5.4) | 34 (100) | |
| Respiratory failure stage, n (%) | | | | <0.0001 |
| 0 | 43 (17.9) | 41 (20) | 2 (5.9) | |
| 1 | 132 (55.2) | 132 (64.4) | 0 | |
| 2 | 22 (9.2) | 22 (10.7) | 0 | |
| 3 | 42 (17.6) | 10 (4.9) | 32 (94.1) | |
| **Comorbidities** | | | | |
| Diabetes, n (%) | 53 (22.2) | 43 (20.98) | 10 (29.4) | 0.27 |
| HbA1c>6.5%, n (%) | 47 (50) | 37 (45.7) | 10 (76.9) | 0.036 |
| HTN, n (%) | 126 (52.7) | 112 (54.6) | 14 (41.2) | 0.14 |
| Cardiovascular disease, n (%) | 69 (28.3) | 61 (29.8) | 8 (23.5) | 0.46 |
| Previous cardiac surgery, n (%) | 55 (23.0) | 44 (21.5) | 11 (32.4) | 0.16 |
| **Radiological lung findings** | | | | |
| Maximum % lung damage (CT), % (Median [IQR]) | 40 [20–60] | 30 [20–50] | 60 [40–80] | 0.0004 |
| CT stage, n (%) | | | | 0.008 |
| 1-stage (<25% damage) | 55 (28.6) | 55 (31.1) | 0 (0) | |
| 2-stage (25–49% damage) | 80 (41.7) | 74 (41.8) | 6 (40) | |
| 3-stage (50–74% damage) | 40 (20.8) | 35 (19.8) | 6 (33.3) | |
| 4-stage (≥75% damage) | 17 (8.9 | 13 (7.3) | 4 (26.7 | |
| Polisegmental features, n (%) | 191 (79.9) | 158 (77.1) | 33 (97.1) | 0.007 |
| Bilateral opacities, n (%) | 199 (93.4) | 166 (92.2) | 33 (100) | 0.097 |
| Pulmonary fibrosis, n (%) | 16 (6.7) | 14 (6.8) | 2 (5.9) | 0.84 |
| **Laboratory findings at admission** | | | | |
| Urea mg/dl (median [IQR]) | 33.3 [25.1–43.5] | 31.6 [24.7–40.6] | 55.7 [37.8–99.6] | <0.0001 |
| Creatinine mg/dl (median [IQR]) | 0.9 [0.7–1.0] | 0.9 [0.7–1.0] | 1.0 [0.8–1.7] | 0.0025 |
| ALT U/l (median [IQR]) | 30.6 [18.7–50.1] | 29.1 [18.5–50.3] | 35.9 [22.2–50.1] | 0.18 |
| AST U/l (median [IQR]) | 33.1 [22.1–49.7] | 31.9 [21.9–48] | 45.4 [30.7–66] | 0.0034 |
| Total bilirubin mg/dl (median [IQR]) | 0.6 [0.4–0.8] | 0.5 [0.40–0.8] | 0.8 [0.6–1.2] | <0.0001 |

*(Continued)*

**Table 1.** (Continued)

| Characteristic | Total (n = 239) | Survived (n = 205) | Deceased (n = 34) | p value |
|---|---|---|---|---|
| Total protein g/dl (mean ± SD) | 6.6±0.7 | 6.7±0.6 | 5.9±0.9 | <0.0001 |
| Albumin g/dl (mean ± SD) | 3.5±0.7 | 3.7±0.6 | 2.8±0.6 | <0.0001 |
| LDH, U/L (median [IQR]) | 255 [190–342] | 247 [190–330.6] | 370 [318–556] | 0.0064 |
| CRP mg/dl (median [IQR]) | 3.5 [0.9–9.1] | 2.9 [0.6–7.8] | 12.8 [7.4–23.8] | <0.0001 |
| HbA1c % (mean ± SD) | 7.1±1.8 | 6.99±1.8 | 7.4±1.2 | 0.42 |
| D-dimer, ng/ml (median [IQR]) | 0.6 [0.3–1.2] | 0.6 [0.3–0.9] | 2.9 [1.0–7.7] | <0.0001 |
| Fibrinogen g/l (median [IQR]) | 4.2 [3.2–5.3] | 4.1 [3.3–5.0] | 5.1 [3.2–6.6] | 0.043 |
| INR (median [IQR]) | 0.96 [0.9–1.1] | 0.9 (0.9–1.0) | 1.1 {1.0–1.3} | <0.0001 |
| aPTT seconds (median [IQR]) | 37.2 [32.8–42] | 36.1 [32.5–40.7] | 43.6 [38.4–68.1] | <0.0001 |
| Ferritin µg/l (median [IQR]) | 335.5 [185.7–779.4] | 292.6 [169.2–651.3] | 1047.1 [674.8–1483] | <0.0001 |
| Positive PCR for SARS-CoV-2, n (%) | 151 (63.2) | 122 (59.5) | 29 (85.3) | 0.004 |
| **Treatment** | | | | |
| Supplemental oxygen, n (%) | 113 (47.3) | 110 (53.7) | 3 (8.8) | <0.0001 |
| Non-invasive mechanical ventilation, n (%) | 5 (2.1) | 3 (1.5) | 2 (5.9) | 0.095 |
| Invasive mechanical ventilation, n (%) | 33 (13.8) | 2 (0.98) | 31 (91.2) | <0.0001 |
| ECMO, n (%) | 14 (5.9) | 2 (0.98) | 12 (35.3) | <0.0001 |
| Renal replacement therapy, n (%) | 27 (11.3) | 5 (2.44) | 22 (64.71) | <0.0001 |
| **Medications** | | | | |
| *Antivirals (total), n (%)* | 50 (20.92) | 27 (13.2) | 23 (67.6) | <0.0001 |
| *Antibiotics (total), n (%)* | 226 (94.6) | 194 (94.6) | 32 (94.1) | 0.90 |
| *Steroids, n (%)* | 188 (78.7) | 168 (81.9) | 20 (58.8) | 0.002 |
| *Anticoagulation (Oral), n (%)* | 27 (11.3) | 27 (13.2) | 0 (0) | 0.025 |
| *Anticoagulation (IV), n (%)* | 201 (84.1) | 174 (84.9) | 27 (79.4) | 0.42 |
| *Aspirin, n (%)* | 57 (23.8) | 54 (26.3) | 3 (8.8) | 0.026 |
| **Complications/outcome** | | | | |
| AKI, n (%) | 56 (24.4) | 27 (13.2) | 29 (85.3) | <0.0001 |
| Direct ICU admission, n (%) | 26 (10.9) | 0 | 26 (76.4) | <0.0001 |
| Hospital stay duration, day median [IQR] | 8 (6–12) | 7 (6–11) | 13 (6–21) | 0.0032 |

PCR, polymerase chain reaction; BMI, body mass index; IQR, interquartile range; HbA1, Hemoglobin Subunit Alpha 1; HTN, Hypertension; ALT U/l, Alanine aminotransferase; LDH, U/L, lactate dehydrogenase; CRP, C-reactive protein, INR, International normalized ratio; CT, computerized tomography; AST U/l, Aspartate aminotransferase; aPTT, Activated partial thromboplastin time; ECMO, extracorporeal membrane oxygenation; ICU, Intensive care unit; AKI, acute kidney injury.

After adjustments in different models (Table 3), age (OR 1.20; 95% CI 1.01–1.43; p = 0.035), respiratory rate (OR 1.38; 95% CI 1.07–1.77; p = 0.013) and CRP (OR 1.39; 95% CI 1.04–1.87; p = 0.026) remained as independent factors associated with in-hospital mortality.

## Discussion

We found that age, respiratory rate and CRP were independent predictors of mortality in our cohort. According to WHO, -"A death due to COVID-19 is defined for surveillance purposes as a death resulting from a clinically compatible illness, in a probable or confirmed COVID-19 case, unless there is a clear alternative cause of death that cannot be related to COVID disease (e.g. trauma). There should be no period of complete recovery from COVID-19 between illness and death. A death due to COVID-19 may not be attributed to another disease (e.g.

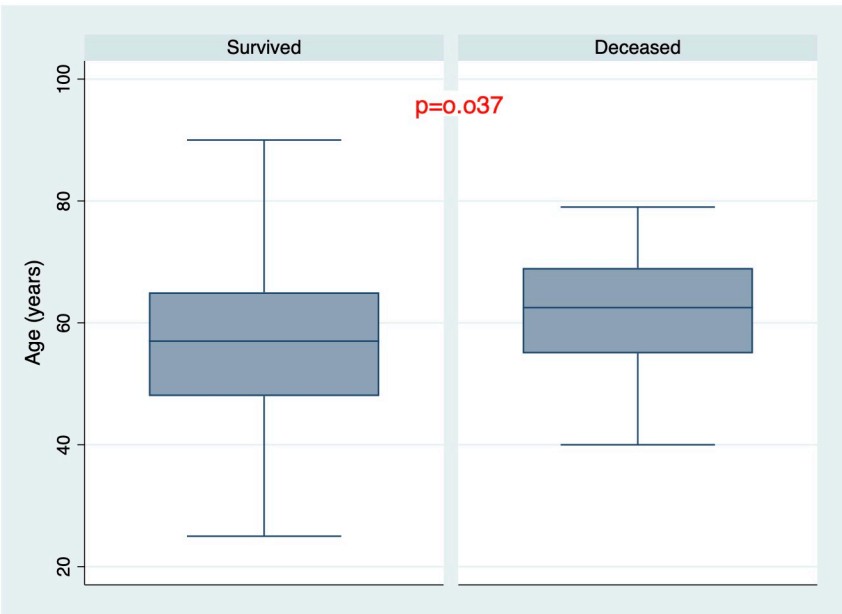

**Fig 1. Age distribution among survived and deceased patients.**

cancer) and should be counted independently of preexisting conditions that are suspected of triggering a severe course of COVID-19" [10].

In the face of the COVID-19 pandemic, a number of tertiary hospitals around the globe had to re-organize into the COVID-centers to help with the overflow in the infectious disease centers [11–13]. Similarly, in Kazakhstan, during the peak of pandemic various hospitals were re-organized into COVID centers. One of them was the National Research Cardiac Surgery

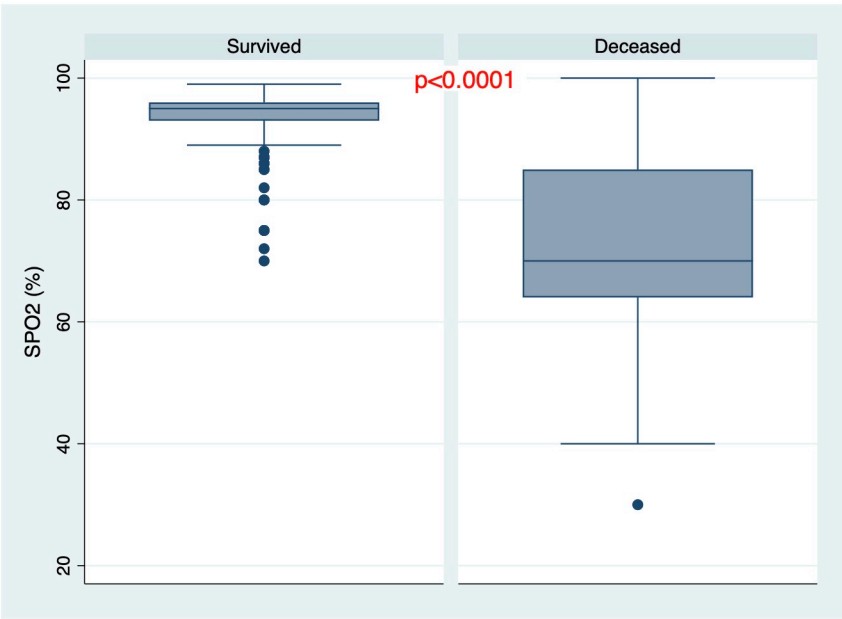

**Fig 2. Saturation percentage among survived and deceased patients.**

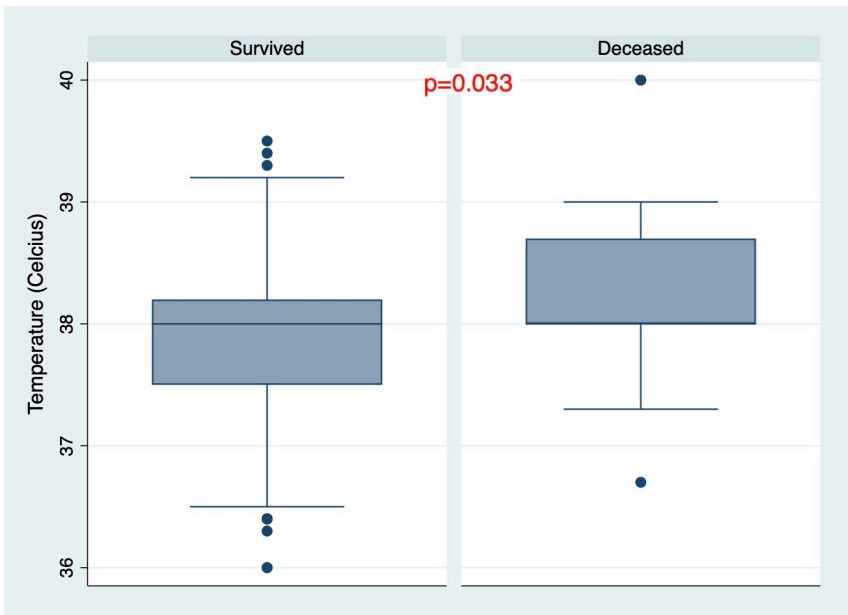

**Fig 3. Temperature distribution among survived and deceased patients.**

Center. Often the most severe cases and critically ill patients were referred to our hospital, which has sufficient experience and equipment to provide advanced floor and ICU care. This could explain why the case fatality rate in our study 3–4 folds was higher than nationwide or worldwide COVID-19 fatality rate of 3–4% [2,13].

Previous studies showed that older age is associated with increased risk of mortality [4,14]. This is consistent with findings from our study, where older age was found to be an

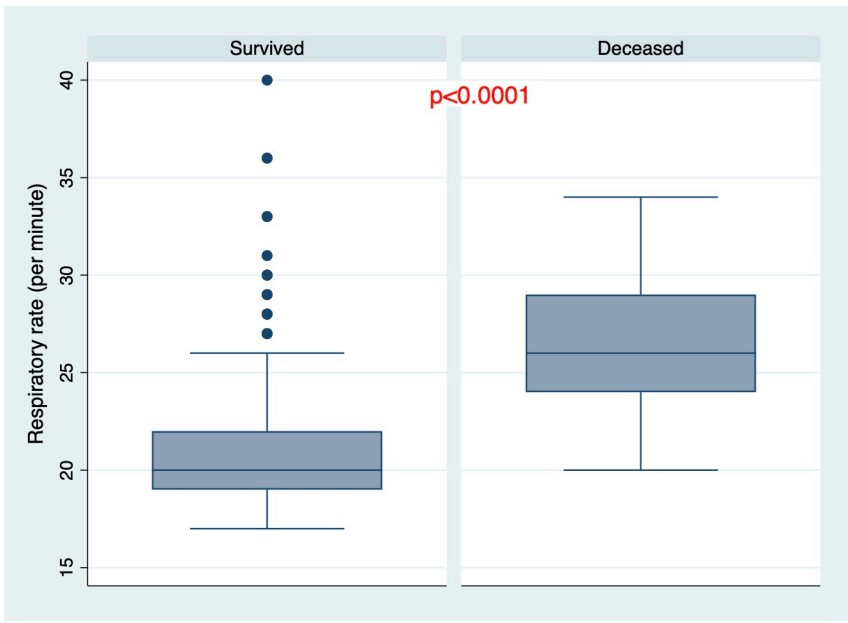

**Fig 4. Respiratory rate distribution among survived and deceased patients.**

**Table 2. The association between demographics, clinical, laboratory, treatment related variable and risk of in-hospital death using unadjusted logistic regression.**

| Characteristic | number of observations / number of events | Univariate OR (95% CI) | p value |
|---|---|---|---|
| Sex | 239 / 34 | 2.3 (0.99–5.33) | 0.052 |
| Age | 239 / 34 | 1.03 (1.00–1.06) | **0.039** |
| Age$^{+5}$ | | 1.17 (1.01–1.36) | |
| BMI | 212 / 25 | 1.05 (0.98–1.14) | 0.16 |
| Diabetes | 239 / 34 | 1.57 (0.70–3.53) | 0.28 |
| HbA1c > 6.5% | 94 / 13 | 3.96 (1.02–15.48) | **0.048** |
| HTN | 239 / 34 | 0.58 (0.28–1.21) | 0.15 |
| Cardiovascular disease | 239 / 34 | 0.73 (0.31–1.69) | 0.46 |
| Previous cardiac surgery | 239 / 34 | 1.75 (0.79–3.86) | 0.17 |
| Day of disease at admission | 239 / 34 | 1.06 (1.01–1.11) | **0.026** |
| Day$^{+2}$ | | 1.12 (1.013–1.23) | |
| Day$^{+5}$ | | 1.32 (1.034–1.68) | |
| Day$^{+7}$ | | 1.47 (1.048–2.06) | |
| Day$^{+10}$ | | 1.73 (1.069–2.81) | |
| Cough | 239 / 34 | 0.71 (0.34–1.47) | 0.36 |
| Temperature | 220 / 23 | 2.06 (1.10–3.83) | **0.023** |
| Temperature$^{+2}$ | | 4.23 (1.22–14.68) | |
| Respiratory rate | 217 / 15 | 1.23 (1.11–1.37) | **<0.0001** |
| Respiratory rate$^{+5}$ | | 2.84 (1.69–4.78) | |
| Respiratory rate$^{+7}$ | | 4.31 (2.08–8.94) | |
| Saturation | 140 / 25 | 0.84 (0.79–0.89) | **<0.0001** |
| Saturation$^{-3}$ | | 1.66 (1.39–1.99) | |
| Saturation$^{-5}$ | | 2.33 (1.72–3.15) | |
| Saturation$^{-7}$ | | 3.26 (2.14–4.97) | |
| Polisegmental CT | 239 / 34 | 9.82 (1.31–73.70) | **0.026** |
| PCR test | 239 / 34 | 3.95 (1.47–10.61) | **0.007** |
| Neutrophils | 239 / 34 | 1.26 (1.16–1.37) | **<0.0001** |
| Neutrophils$^{+5}$ | | 3.22 (2.12–4.88) | |
| Lymphocytes | 238 / 33 | 0.91 (0.87–0.96) | **<0.0001** |
| Lymphocytes$^{-5}$ | | 1.58 (1.26–1.99) | |
| NLR | 238 / 33 | 1.29 (1.11–1.50) | **0.001** |
| NLR$^{+5}$ | | 3.61 (1.69–7.69) | |
| WBC | 238 / 33 | 1.27 (1.16–1.37) | **<0.0001** |
| WBC$^{+5}$ | | 3.18 (2.12–4.78) | |
| PLT | 238 / 33 | 1 (1–1.002) | 0.56 |
| RBC | 238 / 33 | 0.44 (0.25–0.77) | **0.004** |
| RBC$^{-2}$ | | 5.26 (1.67–16.5) | |
| HGB | 238 / 33 | 0.97 (0.95–0.99) | **0.001** |
| HGB$^{-5}$ | | 1.16 (1.058–1.26) | |
| HGB$^{-10}$ | | 1.34 (1.12–1.60) | |
| Urea | 238 / 33 | 1.056 (1.035–1.076) | **<0.0001** |
| Urea$^{-5}$ | | 1.31 (1.19–1.44) | |
| Creatinine | 237/33 | 4.17 (1.89–9.17) | **<0.0001** |
| Creatinine$^{+2}$ | | 17.37 (3.59–84.1) | |
| Total protein | 236/33 | 0.17 (0.086–0.33) | **<0.0001** |
| Total protein$^{-2}$ | | 35.11 (9.15–134.7) | |

*(Continued)*

**Table 2.** (Continued)

| Characteristic | number of observations / number of events | Univariate OR (95% CI) | p value |
|---|---|---|---|
| Albumin | 197/33 | 0.12 (0.054–0.27) | **<0.0001** |
| Albumin$^{-2}$ | | 69.46 (13.96–345.49) | |
| Bilirubin | 238/33 | 3.045 (1.64–5.66) | **<0.0001** |
| Bilirubin$^{+5}$ | | 261.89 (11.76–5834.21) | |
| ALT | 237 / 33 | 1.00 (0.99–1.01) | 0.41 |
| AST | 238 / 33 | 1.01 (1.01–1.013) | **0.027** |
| LDH | 195 / 34 | 1.0039 (1.0012–1.01) | **0.005** |
| LDH$^{+50}$ | | 1.22 (1.063–1.39) | |
| CRP | 233 / 31 | 1.18 (1.11–1.24) | **<0.0001** |
| CRP$^{+5}$ | | 2.24 (1.70–2.95) | |
| Ferritin | 226 / 29 | 1.002 (1.0014–1.003) | **<0.0001** |
| Ferritin$^{+50}$ | | 1.12 (1.075–1.17) | |
| Ferritin$^{+100}$ | | 1.26 (1.16–1.36) | |
| GlycHGB | 94 / 13 | 1.13 (0.84–1.54) | 0.42 |
| INR | 234 / 33 | 1.60 (1.02–2.53) | **0.043** |
| aPTT | 238 / 33 | 1.02 (1.01–1.037) | **<0.0001** |
| aPTT$^{+5}$ | | 1.12 (1.055–1.199) | |
| Fibrinogen | 235 / 33 | 1.3 (1.06–1.59) | **0.013** |
| Fibrinogen$^{+5}$ | | 3.69 (1.32–10.31) | |
| D-dimer | 224 / 32 | 1.39 (1.19–1.62) | **<0.0001** |
| D-dimer$^{+5}$ | | 5.13 (2.36–11.14) | |
| Antivirals, | 238/23 | 13.8 (6.04–31.44) | **<0.0001** |
| Antibiotics, | 238/32 | 0.91 (0.19–4.28) | 0.902 |
| Corticosteroids, | 238/20 | 0.31 (0.15–0.68) | **0.003** |

BMI, Body mass index; HbA1c, Hemoglobin Subunit Alpha 1; HTN, Hypertension; CT, computerized tomography; PCR, polymerase chain reaction; NLR, Neutrophil-Lymphocyte Ratio; WBC, white blood cell count; PLT, platelet count; RBC, Red blood cell count; HGB, Hemoglobin; ALT, Alanine aminotransferase; AST, Aspartate aminotransferase; LDH, Lactate dehydrogenase; CRP, C-reactive protein; GlycHGB, glycosylated hemoglobin; INR, International normalized ratio; aPTT, activated partial thromboplastin time.

**Table 3. The association between demographics, clinical, laboratory, treatment related variable and risk of in-hospital death using adjusted logistic regression models.**

| Covariates | Model 1 OR (95% CI) | p value | Model 2 OR (95% CI) | p value | Model 3 OR (95% CI) | p value |
|---|---|---|---|---|---|---|
| Age | 1.07 (1.01–1.13) | 0.016 | 1.09 (1.01–1.19) | 0.037 | 1.2 (1.01–1.43) | 0.035 |
| Sex | 1.14 (0.34–3.78) | 0.827 | 0.65 (0.11–3.72) | 0.63 | 0.13 (0.01–2.20) | 0.16 |
| Disease day | 1.03 (0.95–1.12) | 0.466 | 0.99 (1.03–1.39) | 0.89 | 0.81 (0.62–1) | 0.14 |
| Respiratory rate | 1.22 (1.09–1.37) | <0.001 | 1.20 (1.03–1.39) | 0.018 | 1.38 (1.07–1.77) | 0.013 |
| WBC | - | - | 1.15 (1.01–1.31) | 0.03 | 1.30 (0.96–1.76) | 0.086 |
| Serum creatinine | - | - | 1.3 (0.52–3.27) | 0.57 | 0.80 (0.35–1.80) | 0.59 |
| Total protein | - | - | 0.41 (0.10–1.73 | 0.23 | 0.050 (0.001–1.47) | 0.083 |
| CRP | - | - | 1.13 (1.00–1.28) | 0.043 | 1.39 (1.04–1.87) | 0.026 |
| Fibrinogen | - | - | 0.82 (0.50–1.35) | 0.44 | 0.54 (0.22–1.35) | 0.19 |
| Hemoglobin | - | - | 1.01 (0.96–1.06) | 0.72 | 1.02 (0.97–1.07) | 0.41 |
| Antivirals | - | - | - | - | 508.5 (5.67–45633) | 0.007 |
| Corticosteroid | - | - | - | - | 3.35 (0.17–66) | 0.43 |

WBC, white blood cell count; CRP, C-reactive protein.

independent predictor of mortality from COVID-19. Literature review also showed that male sex was reported to be associated with increased in-hospital mortality as well as ICU admissions [14,15]. In this study, even though non-survivors were more likely to be males, logistic regression failed to reveal significant difference possibly due to a low sample number.

This study found that the day of disease from the onset of symptoms to admission was associated with higher odds of death, which is supported by evidence from previous research [5]. Importantly, we also confirmed that the longer the waiting time, the higher the odds of non-surviving, which emphasizes the importance of early diagnosis, observation and triage of patients in order to prevent unfavorable outcome.

In this study it was founded that reduced oxygen saturation, elevated body temperature, respiratory rate and degree of respiratory failure are associated with higher odds of death. Moreover, respiratory rate was concluded to be an independent predictor of mortality. Consistently, severity of the disease, which is judged upon the abnormality of these clinical symptoms, was also associated with increased odds of death. These findings are consistent with previously reported [2]. It is important to note that these characteristics in general are easily obtained in a hospital or outpatient, and if abnormal should prompt further testing to make a decision regarding hospital admission.

Previous studies report that presence of comorbidities specifically cardiovascular diseases, hypertension and diabetes are associated with increased risk of death from COVID-19 [14,16,17]. The results of this study showed no association between these variables. Importantly, however, this study showed that elevated glycated hemoglobin level was associated with increased odds of death. This could be explained by the presence of uncontrolled and/or previously undiagnosed diabetes in the patients from our cohort. We hence propose that level of glycated hemoglobin should be assessed upon admission for patients at increased risk of diabetes, despite not previously documented diagnosis. Although limited by missing data, this is an important finding that should be studied more carefully.

This study results support previously reported findings [5,18,19] that the baseline laboratory characteristics could serve as predictors of mortality. Specifically, we concluded that elevated white blood cell count as well as elevated CRP are independent predictors of mortality in our cohort. Other laboratory findings in complete blood count, biochemical, hematological and inflammatory profiles that were found to be more likely abnormal in non-survivors in our cohort are consistent with previously reported data [2,5,18–20].

Although limited by number of patients in the cohort, we showed that non-survivors were more likely to require supplemental oxygen and advanced ICU care with intubation, renal replacement therapy and ECMO. Non-survivors also more often were treated with antivirals. Acute kidney injury was more likely to develop in non-survivors. These are in consistency with previously reported findings [21].

Based on previous studies according to Giovanni et all [24], which is described the work Qin et al. [22] we analyzed markers, in particular, white blood cells and CRP in a cohort of 450 patients with confirmed COVID-19, and found that the severe course of the disease is characterized by an increased level of white blood cells and an increase in C-reactive protein, compared with patients with a moderate course. Similarly, in other works, for example, in Henry et al. It was also concluded in a meta-analysis of 21 studies involving 3377 patients with confirmed COVID19 that patients with a severe course of the disease and fatal outcomes had very high levels of white blood cells, compared with patients with a mild course of the disease and recovered patients. [23,24] What we observed in our patients as well.

This study is limited by the small sample size, missing data for some variables and high proportion of cases not confirmed by PCR. Recent study from Kazakhstan also reported the 85% of SARS-CoV-2 PCR negative cases of pneumonia during the peak of COVID-19 infection

[25]. In addition, the study was conducted in a very specialized tertiary hospital, which limits generalizability of the results. Patients who had diet most like to received antivirals, which was the single option to prescribe them at that time and results indicated it as a significant associated factor with mortality. However, this effect is potential to reverses epidemiology due to treatment bias.

## Conclusion

In conclusion, this is the first study assessing mortality of COVID-19 patients from the tertiary hospital in Kazakhstan. This study describes 14% in-hospital mortality from COVID-19. Many abnormal clinical and laboratory data at admission associated with poor outcome. Age, respiratory rate and CRP were found to be independent predictors of mortality. The findings of this study could help with triage of such patients in order to avoid overload of healthcare facilities and limit involvement of highly specialized centers to cases that are more likely to develop severe disease course. Our finding would help healthcare providers to predict the risk factors associated with high risk of mortality. Further investigations involving large cohorts should be provided to support our findings.

## Author Contributions

**Conceptualization:** Yuriy Pya, Makhabbat Bekbossynova, Abduzhappar Gaipov, Aliya Sailybayeva.

**Data curation:** Makhabbat Bekbossynova, Abduzhappar Gaipov, Ainur Tauekelova, Liya Litvinova.

**Formal analysis:** Abduzhappar Gaipov.

**Investigation:** Yuriy Pya, Makhabbat Bekbossynova, Abduzhappar Gaipov, Timur Lesbekov, Timur Kapyshev, Aidyn Kuanyshbek, Ainur Tauekelova, Liya Litvinova, Aliya Sailybayeva, Ivan Vakhrushev, Antonio Sarria-Santamera.

**Methodology:** Makhabbat Bekbossynova, Abduzhappar Gaipov.

**Project administration:** Yuriy Pya, Makhabbat Bekbossynova, Timur Kapyshev, Ainur Tauekelova, Aliya Sailybayeva.

**Resources:** Abduzhappar Gaipov.

**Supervision:** Yuriy Pya, Makhabbat Bekbossynova, Aliya Sailybayeva, Antonio Sarria-Santamera.

**Writing – review & editing:** Yuriy Pya, Makhabbat Bekbossynova, Abduzhappar Gaipov, Timur Lesbekov, Timur Kapyshev, Aidyn Kuanyshbek, Ainur Tauekelova, Liya Litvinova, Aliya Sailybayeva, Ivan Vakhrushev, Antonio Sarria-Santamera.

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
