## [Decision Letter · Decision Letter 0]

7 Sep 2021

PONE-D-21-25018Mortality predictors of hospitalized patients with COVID-19:Retrospective Cohort Study from Nur-Sultan, KazakhstanPLOS ONE

Dear Dr. Sailybayeva,

Thank you for submitting your manuscript to PLOS ONE. After careful consideration, we feel that it has merit but does not fully meet PLOS ONE’s publication criteria as it currently stands. Therefore, we invite you to submit a revised version of the manuscript that addresses the points raised during the review process.

We look forward to receiving your revised manuscript.

Kind regards,

Muhammad Tarek Abdel Ghafar, M.D

Academic Editor

PLOS ONE

Journal Requirements:

2. During the internal evaluation of the manuscript we have noted a discrepant in the informed consent procedure reported. For instance in the ethics statement on the online submission form, it is stated that informed consent was obtained from participants, however within the manuscript text, it is reported that informed consent was exempt. Please could you provide some further clarification on this.

3. In ethics statement in the manuscript and in the online submission form, please provide additional information about the patient records/samples used in your retrospective study. Specifically, please ensure that you have discussed whether all data/samples were fully anonymized before you accessed them and/or whether the IRB or ethics committee waived the requirement for informed consent. If patients provided informed written consent to have data/samples from their medical records used in research, please include this information.

4. Please update your submission to use the PLOS LaTeX template. The template and more information on our requirements for LaTeX submissions can be found at http://journals.plos.org/plosone/s/latex.

7. PLOS requires an ORCID iD for the corresponding author in Editorial Manager on papers submitted after December 6th, 2016. Please ensure that you have an ORCID iD and that it is validated in Editorial Manager. To do this, go to ‘Update my Information’ (in the upper left-hand corner of the main menu), and click on the Fetch/Validate link next to the ORCID field. This will take you to the ORCID site and allow you to create a new iD or authenticate a pre-existing iD in Editorial Manager. Please see the following video for instructions on linking an ORCID iD to your Editorial Manager account: https://www.youtube.com/watch?v=_xcclfuvtxQ

Reviewers' comments:

Reviewer's Responses to Questions

**Comments to the Author**

1. Is the manuscript technically sound, and do the data support the conclusions?

Reviewer #1: Yes

Reviewer #2: Partly

Reviewer #3: Yes

2. Has the statistical analysis been performed appropriately and rigorously? 

Reviewer #1: Yes

Reviewer #2: Yes

Reviewer #3: Yes

3. Have the authors made all data underlying the findings in their manuscript fully available?

Reviewer #1: Yes

Reviewer #2: Yes

Reviewer #3: Yes

4. Is the manuscript presented in an intelligible fashion and written in standard English?

Reviewer #1: Yes

Reviewer #2: No

Reviewer #3: Yes

5. Review Comments to the Author

Reviewer #1: Mortality predictors of hospitalized patients with COVID-19:Retrospective Cohort Study

from Nur-Sultan, Kazakhstan

The study has merits, and important for understanding the characteristics of patient who are likely to die from COVID-19 and institute steps to avert that. However, I do have some concerns that need to be resolved;

1. Ethical statement: preliminary information section stated that “The study was approved by the Institutional Review Ethics Committee of the National Research Cardiac Surgery Center (#01-97/2021 from 22/04/21). The written consent was obtained”

a. Was wondering how written consent was obtained in a retrospective study

b. However contradicting information was found on, Page 7, lines 177-178 that stated “Therefore this study was exempt from informed consent acquisition process.”

2. Page 4, lines 109- 111; described the inclusion criteria “Diagnosis of COVID-19 was defined as either a positive nasopharyngeal swab for SARS-CoV-2 by reverse transcriptase polymerase chainmreaction (PCR) and/or clinically supported evidence of SARS-CoV-2 in the absence of PCR confirmed test”

a. “The clinically supported evidence of SARS-CoV-2 in the absence of PCR confirmed test” needs further clarity.

i. What clinical criteria was used?

ii. The criteria above suggests that the presence of “clinical symptoms” in a PCR negative person still qualifies the person for classification as COVID-19 case in this study

3. Page 6, lines 132- 134 “……antibiotics (Cefuroxim, Cefazolin, Ceftriaxon, Cefepim. Moxifloxacin, Aztreonam. Meropinem. Levafloxacin. Fluconazole), anticoagulation and steroids were recorded”

a. The phrase needs revising, the multiple full stops need to be checked

b. Are authors suggesting that Fluconazole is an antibiotic

4. Page 6, study definitions: lines 138 – 142 stated the criteria for disease severity:

“Severity of the disease was evaluated based on clinical symptoms in accordance with national clinical guidelines. “The disease severity was stratified as: mild disease – body temperature <38°С, heart rate <90 bpm, respiratory rate <24/min; moderate disease – body temperature 38.1-39°С, heart rate 90-120 bpm, respiratory rate > 24/min; and severe disease – high temperature (over 39°С) with severe symptoms (headache, myalgia, nausea), heart rate > 120 bpm, respiratory rate >28/min.

a. I appreciate your national guidelines but for the purpose or research and uniformity in comparing several studies across many countries, I would advise you use globally acceptable severity classification; eg WHO classification of COVID-19 severity

b. Also, your classification based on temperature and clinical features described as “severe symptoms (headache, myalgia, nausea)” may lead to misclassification bias;

i. The clinical features described as severe symptoms are very subjective depending on the pain threshold of patients etc

ii. Also, it is known that not all patients with COVID-19 report high grade fever, hence using fever and some other subjective symptoms as criteria for measuring severity needs further explanation.

5. Page 7, lines 174-175 reads “The outpatient cards of patients are presented by anonymized database without any identifiable information about patients.”

a. The meaning is not clear

b. Also, “outpatient cards” a little confusing; I though our study participants were “inpatients”

6. Page 8, results section; line 185 states; “151 patients (63.2%) had PCR confirmed COVID-19 diagnosis.

a. The 54 who did not have a COVID-19 diagnosis, what disease condition were there suffering from

b. Since your title, and objective is to describe “Mortality predictors of hospitalized patients with COVID-19: Retrospective Cohort Study from Nur-Sultan, Kazakhstan”

i. I find it difficult to understand why other patients for which a diagnosis of COVID-19 was not confirmed were included.

ii. The symptoms of COVID-19 symptoms are really similar to that of various other infectious diseases, and it is possible most of the 54 without a confirmed PCR diagnosis were actually not COVID-19 cases. Though it was mentioned in the limitations, I feel it is a fundamental issue that affects the estimates made and, that need to be looked at again.

Reviewer #2: General

• The paper needs to be reviewed by native English speaker

Abstract

• Line 46: Do not write numbers in figure at the beginning of any sentence.

• Line 55: Is it discussion or conclusion?

• Line 56 – 58: You need to put recommendations and/or policy/strategic options for your findings. Otherwise, the findings may not have much relevance.

Methods

• Lines 107 – 111: Is it possible to diagnose COVID-19 clinically? It has similar sign and symptoms with other respiratory diseases including common cold. Do you think that it doesn’t affect your findings?

• Describe number of participants, inclusion and exclusion and ethical statement separately. It is not part of the statistical analysis.

• In this part, you should indicate that how did you ascertain mortality due to COVID-19. How could you be sure that the deaths are not due to other medical conditions? Death definition for COVID-19 is required.

Result

• Line 185: Do not write numbers in figure at the beginning of any sentence. Write it in word.

• Lines 189 – 191: The narration here looks conclusion or discussion. Here, the author should narrate the actual value (avoid using of more, most, more likely….)

• In Table 1: Under the variable gender, include also other categories like female or other categories if any like you for obesity in the same table. This table also need footnote, there so many abbreviations and symbols in the table. These should be described/stated in footnote.

• Lines 194 – 200: Is it the appropriate place for the list of figure titles?

• Lines 202 – 208: Similar comment with lines 189 – 191.

• Tables 2 & 3 need footnote as well.

• All figures also need footnotes and abbreviations in the lend need to stated there.

Discussion

• I don’t think that the first sentence of the discussion is relevant.

• Lines 239 – 244: This is stated also in the introduction part. I think these statements are better in the introduction. These are not your findings/I haven’t found them in the result part. Therefore, my recommendation here is to remove from the discussion and keep in the introduction part.

• Lines 275 – 280: The authors stated that the derangements of laboratory findings are predictors of mortality. This is very interesting. However, you need to explain why, what are the implications here.

• Lines 291 – 297: Conclusion should be a separate heading and, in this part, put your recommendations and policy/strategic directions other than putting the importance of the findings.

Reviewer #3: 1. Line 229 - 231 stated that there are 3 independent factors associated with in-hospital mortality: ag, respiratory rate and CRP. In Table 3, Antiviral has p-value 0.007. Please clarify why it is not considered as one of the independent factors to in-hospital mortality.

2. Are those with diagnosis not confirmed with PCR taken during initial presentation had their PCR re-taken during the course of their admission? Is there a reason why these patients considered to be Covid-19 when there are no positive result?

6. PLOS authors have the option to publish the peer review history of their article (what does this mean?). If published, this will include your full peer review and any attached files.

Reviewer #1: No

Reviewer #2: No

Reviewer #3: **Yes: **Nurul Huda Ahmad

---

## [Author Response · Author response to Decision Letter 0]

7 Oct 2021

Answers to the review Comments to the Author

Reviewer #1:

2. During the internal evaluation of the manuscript we have noted a discrepant in the informed consent procedure reported. For instance in the ethics statement on the online submission form, it is stated that informed consent was obtained from participants, however within the manuscript text, it is reported that informed consent was exempt. Please could you provide some further clarification on this.

- Ethical statement: preliminary information section stated that “The study was approved by the Institutional Review Ethics Committee of the National Research Cardiac Surgery Center (#01-97/2021 from 22/04/21). The written consent was obtained”

-Answer: We agree with your comments and sorry for this technical mistake. Since the study is retrospective and exemption from informed consent approved by ethic committee decision. 

- 2. Page 4, lines 109- 111; described the inclusion criteria “Diagnosis of COVID-19 was defined as either a positive nasopharyngeal swab for SARS-CoV-2 by reverse transcriptase polymerase chainmreaction (PCR) and/or clinically supported evidence of SARS-CoV-2 in the absence of PCR confirmed test”

a. “The clinically supported evidence of SARS-CoV-2 in the absence of PCR confirmed test” needs further clarity.

I. What clinical criteria was used?

-Answer: The main clinical criterion is: COVID-19-associated pneumonia. Based on the established diagnosis of pneumonia according to CT data, where the leading signs in the early stages of the disease were foci of "frosted glass", multifocality of lung damage, edema of the interalveolar pulmonary interstitium, which was the difference between COVID-19-associated pneumonia from that of another etiology.

In addition to pneumonia, the following clinical criteria were taken into account: close contact with infected patients, clinical signs of COVID-19 which are available elsewhere.

II. The criteria above suggests that the presence of “clinical symptoms” in a PCR negative person still qualifies the person for classification as COVID-19 case in this study

- Answer: Patients with a negative PCR test, but with clinical signs of COVID-19, such as pneumonia (confirmed by chest CT), were selected for the study in accordance with the WHO International Classification. The code U07. 2 is COVID-19, where the COVID-19 virus has not been identified, diagnosed clinically or epidemiologically, but laboratory studies are inconclusive or unavailable. https://icd.who.int/browse10/2019/en#/U07.2

3. Page 6, lines 132- 134 “……antibiotics (Cefuroxim, Cefazolin, Ceftriaxon, Cefepim. Moxifloxacin, Aztreonam. Meropinem. Levafloxacin. Fluconazole), anticoagulation and steroids were recorded”

a. The phrase needs revising, the multiple full stops need to be checked

- Answer: done!

b. Are authors suggesting that Fluconazole is an antibiotic

- Answer: We agree that fluconazole is not an antibiotic, this is an antimicrobial drug. Therefore, we have corrected this point.

-Page 6, study definitions: lines 138 – 142 stated the criteria for disease severity:

“Severity of the disease was evaluated based on clinical symptoms in accordance with national clinical guidelines. “The disease severity was stratified as: mild disease – body temperature <38°С, heart rate <90 bpm, respiratory rate <24/min; moderate disease – body temperature 38.1-39°С, heart rate 90-120 bpm, respiratory rate > 24/min; and severe disease – high temperature (over 39°С) with severe symptoms (headache, myalgia, nausea), heart rate > 120 bpm, respiratory rate >28/min.

a. I appreciate your national guidelines but for the purpose or research and uniformity in comparing several studies across many countries, I would advise you use globally acceptable severity classification; eg WHO classification of COVID-19 severity

Answer: Actually we have followed the WHO classification of the disease severity, but it seems our definition in the manuscript was not clear. Our national guidelines were also fully adapted and written on the basis of Clinical management, WHO, Living guidance, 25 January 2021. https://www.who.int/publications/i/item/WHO-2019-nCoV-clinical-2021-1 ). 

Also, your classification based on temperature and clinical features described as “severe symptoms (headache, myalgia, nausea)” may lead to misclassification bias;

Answer: We agree with you, perhaps these symptoms can be attributed to general non-specific symptoms. But in the case of our study, we assessed the severity of the patients ' condition in conjunction with objective data such as the development of COVID-associated pneumonia according to CT results, signs of respiratory inaccuracy (decreased oxygen saturation) as well as laboratory biomarkers of severity: increases in the level of C reactive protein, ferritin, d-dimmer. 

The clinical features described as severe symptoms are very subjective depending on the pain threshold of patients etc

Answer: The severity of the patients ' condition is assessed by objective signs: laboratory data, instrumental data (CT and X-ray), the degree of lung damage, biochemical indicators, the presence of comorbid diseases.

i. Also, it is known that not all patients with COVID-19 report high grade fever, hence using fever and some other subjective symptoms as criteria for measuring severity needs further explanation.

Answer: The body temperature of patients was measured starting from the emergency room and in the department in dynamics several times by medical personnel. 

5. Page 7, lines 174-175 reads “The outpatient cards of patients are presented by anonymized database without any identifiable information about patients.”

a. The meaning is not clear

Answer: During the data collection, all personal information of patients is encoded and the data is depersonalized, so we protect the rights of patients and do not disclose their personal information. We have little clarified this sentence in the manuscript

b. Also, “outpatient cards” a little confusing; I though our study participants were “inpatients”

Answer: We agree with the remark, we changed it to medical records (medical report). All the patients were on inpatient treatment. We apologize for this mistyping

6. Page 8, results section; line 185 states; “151 patients (63.2%) had PCR confirmed COVID-19 diagnosis.

a. The 54 who did not have a COVID-19 diagnosis, what disease condition were there suffering from

Answer: Actually, all patients had COVID-19 disease with or without PCR positivity. From the cohort, 88 patients (not 54!) presented with PCR negative COVID-19 disease (virus not identified coronavirus pneumonia).

b. Since your title, and objective is to describe “Mortality predictors of hospitalized patients with COVID-19: Retrospective Cohort Study from Nur-Sultan, Kazakhstan”

i. I find it difficult to understand why other patients for which a diagnosis of COVID-19 was not confirmed were included.

Answer: We apologize for this confusion. The text states that the diagnosis of COVID-19 has not been confirmed on the basis of PCR for the rest of the patients, but the diagnosis is made on such a clinical criterion as pneumonia. In clinical practice, there are many cases when PCR is not convincing in patients and therefore the WHO classification has the ICD-10 code U07.2 (COVID-19, virus not identified).

ii. The symptoms of COVID-19 symptoms are really similar to that of various other infectious diseases, and it is possible most of the 54 without a confirmed PCR diagnosis were actually not COVID-19 cases. Though it was mentioned in the limitations, I feel it is a fundamental issue that affects the estimates made and, that need to be looked at again.

Answer: We agree with you that the symptoms of COVID-19 are similar to other respiratory infections. But as we indicated above, in 88 patients (not 54 patients) who had negative PCR results, the diagnosis of COVID-19 was established on the basis of CT- characteristic of COVID-19 pneumonia and other biomarkers such as: increased CRP, ferritin, d-dimer. Our experience was based on WHO clinical guidelines and other literature data.

Reviewer #2: General

• The paper needs to be reviewed by native English speaker

Answer: The manuscript was revised by native English speaker and some correction were made through the text

Abstract

• Line 46: Do not write numbers in figure at the beginning of any sentence.

Answer: The remark is eliminated

• Line 55: Is it discussion or conclusion?

Answer: The remark is eliminated. This section means the conclusion. 

• Line 56 – 58: You need to put recommendations and/or policy/strategic options for your findings. Otherwise, the findings may not have much relevance.

Answer: we thank reviewer for this recommendation. We provided the suggested recommendation in conclusions . “Our finding would help healthcare providers to predict the risk factors associated with high risk of mortality. Further investigations involving large cohorts should be provided to support our findings.”

Methods

• Lines 107 – 111: Is it possible to diagnose COVID-19 clinically? It has similar sign and symptoms with other respiratory diseases including common cold. Do you think that it doesn’t affect your findings?

Answer: To confirm the diagnosis, we used Clinical management, WHO, Living guidance, 25 January 2021 ( https://www.who.int/publications/i/item/WHO-2019-nCoV-clinical-2021-1) Similar comments responded for Reviewer 1 (please see above)

-Describe number of participants, inclusion and exclusion and ethical statement separately. It is not part of the statistical analysis. 

Answer: No patients were excluded from the study, all 239 patients who were admitted to the COVID-19 department during the June 16th – August 30th period were included to the study. We mentioned this in the section “Study Design and Participants”. The ethical statement transferred to the appropriate section.

-In this part, you should indicate that how did you ascertain mortality due to COVID-19. How could you be sure that the deaths are not due to other medical conditions? Death definition for COVID-19 is required.

Answer: We have established the hospital mortality rate from COVID-19 according to the generally accepted formula: the number of patients who died from COVID-19 x 100/ the number of patients discharged with COVID-19. We are sure that the death occurred from COVID-19, and not from other diseases, since the hospital conducts a clinical analysis of fatal cases together with doctors and the department of treatment quality and patient safety. Also, every patient hospitalized with a diagnosis of COVID-19 was thoroughly examined according to WHO international recommendations.

According to “International guidelines for certification and classification (coding) of covid-19 as cause of death” from WHO /https://www.who.int/classifications/icd/Guidelines_Cause_of_Death_COVID-19.pdf/

A death due to COVID-19 is defined for surveillance purposes as a death resulting from a clinically compatible illness, in a probable or confirmed COVID-19 case, unless there is a clear alternative cause of death that cannot be related to COVID disease (e.g. trauma). There should be no period of complete recovery from COVID-19 between illness and death. A death due to COVID-19 may not be attributed to another disease (e.g. cancer) and should be counted independently of preexisting conditions that are suspected of triggering a severe course of COVID-19.

Result

• Line 185: Do not write numbers in figure at the beginning of any sentence. Write it in word.

Answer: The remark is eliminated

• Lines 189 – 191: The narration here looks conclusion or discussion. Here, the author should narrate the actual value (avoid using of more, most, more likely….) 

Answer: The remark is eliminated

• In Table 1: Under the variable gender, include also other categories like female or other categories if any like you for obesity in the same table. This table also need footnote, there so many abbreviations and symbols in the table. These should be described/stated in footnote.

Answer: comments and suggestions were made

• Lines 194 – 200: Is it the appropriate place for the list of figure titles?

Answer: This is written according to the requirements of the PlosOne guide for authors. Below we give excerpts from the instructions to the authors

«Figures

Do not include figures in the main manuscript file. Each figure must be prepared and submitted as an individual file. 

Figure captions

Figure captions must be inserted in the text of the manuscript, immediately following the paragraph in which the figure is first cited (read order). Do not include captions as part of the figure files themselves or submit them in a separate document.

At a minimum, include the following in your figure captions:

A figure label with Arabic numerals, and “Figure” abbreviated to “Fig” (e.g. Fig 1, Fig 2, Fig 3, etc). Match the label of your figure with the name of the file uploaded at submission (e.g. a figure citation of “Fig 1” must refer to a figure file named “Fig1.tif”).

A concise, descriptive title

The caption may also include a legend as needed.

Cite figures in ascending numeric order at first appearance in the manuscript file.»

• Lines 202 – 208: Similar comment with lines 189 – 191.

Answer: This is written according to the requirements of the PlosOne guide for authors

• Tables 2 & 3 need footnote as well.

Answer: The footnote is put under the tables 2 and 3

• All figures also need footnotes and abbreviations in the lend need to stated there.

Answer: It’s unacceptable according to the requirements of the PlosOne guide for authors

Discussion

• I don’t think that the first sentence of the discussion is relevant.

Answer: We have removed this sentence

• Lines 239 – 244: This is stated also in the introduction part. I think these statements are better in the introduction. These are not your findings/I haven’t found them in the result part. Therefore, my recommendation here is to remove from the discussion and keep in the introduction part.

Answer: We removed these lines from Discussion part and inserted to the Introduction

• Lines 275 – 280: The authors stated that the derangements of laboratory findings are predictors of mortality. This is very interesting. However, you need to explain why, what are the implications here.

Answer: Based on previous studies, for example, Qin et al. We analyzed markers, in particular, white blood cells and CRP in a cohort of 450 patients with confirmed COVID-19, and found that the severe course of the disease is characterized by an increased level of white blood cells and an increase in C-reactive protein, compared with patients with a moderate course. Similarly, in other works, for example, in Henry et al. It was also concluded in a meta-analysis of 21 studies involving 3377 patients with confirmed COVID19 that patients with a severe course of the disease and fatal outcomes had very high levels of white blood cells, compared with patients with a mild course of the disease and recovered patients. What we observed in our patients as well

Source: Biomarkers associated with COVID-19 disease progression Giovanni Ponti, Monia Maccaferri, Cristel Ruini, Aldo Tomasi, Tomris Ozben Crit Rev Clin Lab Sci. 2020 : 1–11. Published online 2020 Jun 5. doi: 10.1080/10408363.2020.1770685 PMCID: PMC7284147

• Lines 291 – 297: Conclusion should be a separate heading and, in this part, put your recommendations and policy/strategic directions other than putting the importance of the findings. 

Answer: We provided the recommendation policy in conclusions. (Our finding would help healthcare providers to predict the risk factors associated with high risk of mortality. Further investigations involving large cohorts should be provided to support our findings)

Reviewer #3: 

1. Line 229 - 231 stated that there are 3 independent factors associated with in-hospital mortality: ag, respiratory rate and CRP. In Table 3, Antiviral has p-value 0.007. Please clarify why it is not considered as one of the independent factors to in-hospital mortality. Answer: because of reverse epidemiology effect (possible to treatment bias) we could not comment on this. All patients who were critically ill had received antivirals (it was single option to prescribe them). This is resulted that most of the critically ill patients died who received antivirals. We provided this information in the limitations.

2. Are those with diagnosis not confirmed with PCR taken during initial presentation had their PCR re-taken during the course of their admission? Is there a reason why these patients considered to be Covid-19 when there are no positive result?

Answer: The PCR test was performed twice. Patients with a negative PCR test, but with clinical signs of COVID-19, such as pneumonia, were selected for the study in accordance with the WHO International Classification. The code U07. 2 is COVID-19, where the COVID-19 virus has not been identified, diagnosed clinically or epidemiologically, but laboratory studies are inconclusive or unavailable. https://icd.who.int/browse10/2019/en#/U07.2

---

## [Decision Letter · Decision Letter 1]

21 Oct 2021

PONE-D-21-25018R1

Mortality predictors of hospitalized patients with COVID-19:Retrospective Cohort Study from Nur-Sultan, Kazakhstan

PLOS ONE

Dear Dr. Sailybayeva,

Thank you for submitting your manuscript to PLOS ONE. After careful consideration, we feel that it has merit but does not fully meet PLOS ONE’s publication criteria as it currently stands. Therefore, we invite you to submit a revised version of the manuscript that addresses the points raised during the review process.

A marked-up copy of your manuscript that highlights changes made to the original version. You should upload this as a separate file labeled 'Revised Manuscript with Track Changes'.An unmarked version of your revised paper without tracked changes. You should upload this as a separate file labeled 'Manuscript'.If applicable, we recommend that you deposit your laboratory protocols in protocols.io to enhance the reproducibility of your results. Protocols.io assigns your protocol its own identifier (DOI) so that it can be cited independently in the future. For instructions see: https://journals.plos.org/plosone/s/submission-guidelines#loc-laboratory-protocols. Additionally, PLOS ONE offers an option for publishing peer-reviewed Lab Protocol articles, which describe protocols hosted on protocols.io. Read more information on sharing protocols at https://plos.org/protocols?utm_medium=editorial-email&utm_source=authorletters&utm_campaign=protocols.

We look forward to receiving your revised manuscript.

Kind regards,

Muhammad Tarek Abdel Ghafar, M.D

Academic Editor

PLOS ONE

Journal Requirements:

Additional Editor Comments (if provided):

1- Please check and ensure that all abbreviations in the text are fully defined in full terms before they are first used in the text.

2- Figures 1, 2, 3, and 4 should be presented as bar graphs with error bars.

Reviewers' comments:

Reviewer's Responses to Questions

**Comments to the Author**

1. If the authors have adequately addressed your comments raised in a previous round of review and you feel that this manuscript is now acceptable for publication, you may indicate that here to bypass the “Comments to the Author” section, enter your conflict of interest statement in the “Confidential to Editor” section, and submit your "Accept" recommendation.

Reviewer #1: All comments have been addressed

Reviewer #2: All comments have been addressed

Reviewer #3: All comments have been addressed

2. Is the manuscript technically sound, and do the data support the conclusions?

Reviewer #1: Yes

Reviewer #2: Yes

Reviewer #3: Yes

3. Has the statistical analysis been performed appropriately and rigorously? 

Reviewer #1: Yes

Reviewer #2: Yes

Reviewer #3: Yes

4. Have the authors made all data underlying the findings in their manuscript fully available?

Reviewer #1: Yes

Reviewer #2: (No Response)

Reviewer #3: Yes

5. Is the manuscript presented in an intelligible fashion and written in standard English?

Reviewer #1: Yes

Reviewer #2: Yes

Reviewer #3: Yes

6. Review Comments to the Author

Reviewer #1: The authors have addressed all my concerns raised in the first round of the review. I do not have any further concerns.

Reviewer #2: (No Response)

Reviewer #3: (No Response)

7. PLOS authors have the option to publish the peer review history of their article (what does this mean?). If published, this will include your full peer review and any attached files.

Reviewer #1: No

Reviewer #2: No

Reviewer #3: **Yes: **Nurul Huda Ahmad

---

## [Author Response · Author response to Decision Letter 1]

14 Nov 2021

- All abbreviations in the text are fully defined in full terms before they are first used in the text.

- Figures 1, 2, 3, and 4 are presented as bar graphs with error bars

---

## [Editor Report · Decision Letter 2]

22 Nov 2021

PONE-D-21-25018R2Mortality predictors of hospitalized patients with COVID-19:Retrospective Cohort Study from Nur-Sultan, KazakhstanPLOS ONE

Dear Dr. Sailybayeva,

Thank you for submitting your manuscript to PLOS ONE. After careful consideration, we feel that it has merit but does not fully meet PLOS ONE’s publication criteria as it currently stands. Therefore, we invite you to submit a revised version of the manuscript that addresses the points raised during the review process. Please submit your revised manuscript by Jan 06 2022 11:59PM. If you will need more time than this to complete your revisions, please reply to this message or contact the journal office at plosone@plos.org. Please include the following items when submitting your revised manuscript:A rebuttal letter that responds to each point raised by the academic editor and reviewer(s). You should upload this letter as a separate file labeled 'Response to Reviewers'.A marked-up copy of your manuscript that highlights changes made to the original version. You should upload this as a separate file labeled 'Revised Manuscript with Track Changes'.An unmarked version of your revised paper without tracked changes. You should upload this as a separate file labeled 'Manuscript'.If applicable, we recommend that you deposit your laboratory protocols in protocols.io to enhance the reproducibility of your results. Protocols.io assigns your protocol its own identifier (DOI) so that it can be cited independently in the future. For instructions see: https://journals.plos.org/plosone/s/submission-guidelines#loc-laboratory-protocols. Additionally, PLOS ONE offers an option for publishing peer-reviewed Lab Protocol articles, which describe protocols hosted on protocols.io. Read more information on sharing protocols at https://plos.org/protocols?utm_medium=editorial-email&utm_source=authorletters&utm_campaign=protocols.

We look forward to receiving your revised manuscript.

Kind regards,

Muhammad Tarek Abdel Ghafar, M.D

Academic Editor

PLOS ONE

Journal Requirements:

Additional Editor Comments:

The type of graphs is mentioned as bar graphs with error which is not correct. The correct is Box plot with whiskers. Please revise and correct.
---

## [Author Response · Author response to Decision Letter 2]

25 Nov 2021

Dear Managing Editor!

We would like to express our deepest gratitude for the time and efforts of the Reviewers and the Editor spent on our manuscript. All the outstanding comments enlightened us, and they will ultimately improve the scientific and practical content of our paper.

We are resubmitting the revised version that incorporates the suggestions made by the and feel that the manuscript is greatly enhanced as a result. 

Point to point answers to the review Comments to the Author

Answer: The retracted article from [Giovanni Ponti, Monia Maccaferri, Cristel Ruini, Aldo Tomasi, Tomris Ozben. Biomarkers associated with COVID-19 disease progression. Crit Rev Clin Lab Sci. 2020 : 1–11. Published online 2020 Jun 5. doi: 10.1080/10408363.2020.1770685] is included with original citations and the Reference list is filled by references from retracted article.

Based on previous studies according to Giovanni et all [24], which is described the work Qin et al. [22] we analyzed markers, in particular, white blood cells and CRP in a cohort of 450 patients with confirmed COVID-19, and found that the severe course of the disease is characterized by an increased level of white blood cells and an increase in C-reactive protein, compared with patients with a moderate course. Similarly, in other works, for example, in Henry et al. It was also concluded in a meta-analysis of 21 studies involving 3377 patients with confirmed COVID19 that patients with a severe course of the disease and fatal outcomes had very high levels of white blood cells, compared with patients with a mild course of the disease and recovered patients. [23, 24] What we observed in our patients as well.

References

22. Qin C, Zhou L, Hu Z, et al. Dysregulation of immune response in patients with COVID-19 in Wuhan, China. Clin Infect Dis. 2020. DOI:10.1093/cid/ciaa248 

23. Henry BM, de Oliveira MHS, Benoit S, et al. Hematologic, biochemical and immune biomarker abnormalities associated with severe illness and mortality in coronavirus disease 2019 (COVID-19): a meta-analysis. Clin Chem Lab Med. 2020. 

24. Giovanni Ponti, Monia Maccaferri, Cristel Ruini, Aldo Tomasi, Tomris Ozben. Biomarkers associated with COVID-19 disease progression. Crit Rev Clin Lab Sci. 2020 : 1–11. Published online 2020 Jun 5. doi: 10.1080/10408363.2020.1770685

-The bar graphs with error are replaced with the Box plot with whisker.

We hope that our manuscript in its current revised format is now suitable for publication in PLOS ONE

We look forward to hearing your decision.

Sincerely, Aliya Sailybaeva

Head of Research department of “National Research Cardiac Surgery Centre” JSC, 010000, Republic of Kazakhstan, Nur-Sultan, Turan Avenue 38, phone number: +77172703153, fax number: +7 (7172) 703 158, dr.alisai@gmail.com, cardiacsurgeryres@gmail.com

---

## [Editor Report · Decision Letter 3]

26 Nov 2021

Mortality predictors of hospitalized patients with COVID-19:Retrospective Cohort Study from Nur-Sultan, Kazakhstan

PONE-D-21-25018R3

Dear Dr. Sailybayeva,

We’re pleased to inform you that your manuscript has been judged scientifically suitable for publication and will be formally accepted for publication once it meets all outstanding technical requirements.

Kind regards,

Muhammad Tarek Abdel Ghafar, M.D

Academic Editor

PLOS ONE
---

## [Editor Report · Acceptance letter]

6 Dec 2021

PONE-D-21-25018R3 

Mortality predictors of hospitalized patients with COVID-19:
Retrospective Cohort Study from Nur-Sultan, Kazakhstan 

Dear Dr. Sailybayeva:

I'm pleased to inform you that your manuscript has been deemed suitable for publication in PLOS ONE. Congratulations! Your manuscript is now with our production department. 

Kind regards, 

on behalf of

Prof Muhammad Tarek Abdel Ghafar 

Academic Editor

PLOS ONE